# A Simplified Method for Considering Achilles Tendon Curvature in the Assessment of Tendon Elongation

**DOI:** 10.3390/s21217387

**Published:** 2021-11-06

**Authors:** Mohamadreza Kharazi, Christos Theodorakis, Falk Mersmann, Adamantios Arampatzis, Sebastian Bohm

**Affiliations:** 1Department of Training and Movement Sciences, Humboldt-Universität zu Berlin, 10115 Berlin, Germany; mohamadreza.kharazi@hu-berlin.de (M.K.); theodchr@hu-berlin.de (C.T.); falk.mersmann@hu-berlin.de (F.M.); a.arampatzis@hu-berlin.de (A.A.); 2Berlin School of Movement Science, 10115 Berlin, Germany

**Keywords:** Achilles tendon, tendon elongation, tendon strain, walking and running, ultrasound

## Abstract

The consideration of the Achilles tendon (AT) curvature is crucial for the precise determination of AT length and strain. We previously established an ultrasound-kinematic-based method to quantify the curvature, using a line of reflective foil skin markers covering the AT from origin to insertion. The current study aimed to simplify the method by reducing the number of markers while maintaining high accuracy. Eleven participants walked (1.4 m/s) and ran (2.5, 3.5 m/s) on a treadmill, and the AT curvature was quantified using reflective foil markers aligned with the AT between the origin on the gastrocnemius myotendinous-junction (tracked by ultrasound) and a marker on the calcaneal insertion. Foil markers were then systematically removed, and the introduced error on the assessment of AT length and strain was calculated. We found a significant main effect of marker number on the measurement error of AT length and strain (p<0.001). Using more than 30% of the full marker-set for walking and 50% for running, the $R^{2}$ of the AT length error saturated, corresponding to average errors of <0.1 mm and <0.15% strain. Therefore, a substantially reduced marker-set, associated with a marginal error, can be recommended for considering the AT curvature in the determination of AT length and strain.

## 1. Introduction

The Achilles tendon (AT) length is an important parameter to assess its mechanical properties and to understand the muscle-tendon interaction within the triceps surae muscle-tendon unit during functional tasks of daily life or athletic performance, such as walking [1], running [2,3], sprinting [4], jumping/landing [5,6,7], and cycling [8,9]. AT length-changes upon loading can also be used to estimate tendon forces and strain energy [10], tendon stiffness [11], the decoupling within the muscle-tendon unit [8], and tendon hysteresis [5]. Furthermore, tendon strain (length-changes normalized to resting length) has been suggested as a predictor for tendon injury risk [12], a regulator of tendon adaptation [3,13], and, in this regard, an important marker for subject-specific tendon exercise prescriptions [14]. Thus, the accurate assessment of AT length is crucial for the field of human tendon biomechanics.

A simple planimetric model [15] is often used for the determination of in vivo AT length but may not reflect the complex in vivo condition [16]. For more sophisticated measures of the AT length, a combination of ultrasound and motion capture has been established previously [5,7,17]. Using this approach, the gastrocnemius medialis muscle-tendon junction (GM MTJ) commonly defines the AT origin and is tracked by ultrasound and transferred to the global coordinate system. The insertion is given by a reflective marker on the calcaneus bone in the same global coordinate system. The AT length is then calculated as the linear distance between the AT origin and insertion. The assumption to this method is that the AT follows a straight line from the origin to insertion and does not change its alignment, i.e., curvature, during the movement tasks. In fact, the AT features a concave curvature that can alter during contractions and functional movements [18,19], calling the straight length method into question. In a previous study [20], we provided an approach to consider the AT curvature for the AT length determination using a line of reflective foil markers (i.e., self-adhesive retro-reflective tape circles with 5 mm in diameter) placed in small spatial intervals on the skin covering the AT from origin to insertion. We showed that when including the instant AT curvature, the AT elongation was significantly lower by 1.2 ± 0.4 mm compared to the AT length measured with the straight AT length method (i.e., no curvature consideration) during a maximal isometric voluntary contraction [21]. The error was found to be substantially higher during more dynamic tasks, e.g., 9.0 ± 5.4 mm during ankle joint rotations (15º dorsiflexion to 30º plantar flexion) [22] and 5.0 ± 1.3 mm during hopping [23]. Together these reports provide evidence that the consideration of the AT curvature is crucial for the precise measurement of AT length and strain. However, using many foil markers on the path of the AT to reconstruct its curvature is experimentally effortful [10]. The increased need for hardware (i.e., number of cameras for high-quality marker tracking) and high time-demand for post-data processing (marker labeling, gap filling and checking for potential foil markers distortion) hamper the practicability and applicability of this method. A simplified AT length assessment using a reduced marker-set may, however, be appropriate to detect the AT length and strain magnitudes and might be of interest when implemented in a diagnostic approach to monitor the mechanical demand of a tendon and control for strain ranges to stimulate adaptation (4.5% to 6.5%) [11,14,24,25]. A reduced marker-set as a straightforward approach could increase the feasibility of the foil marker-based method to reconstruct the AT curvature while maintaining the associated error in a reasonable range. However, it has not been systematically investigated yet how many foil markers are required for a precise AT length determination during locomotion. Furthermore, changes in AT curvature during locomotion and thus the required markers for reconstruction may depend on the gait phase (stance vs. swing), gait mode (walking vs. running), and gait speed (slow vs. fast) because of different contraction intensities and ankle joint range of motions that are involved.

The purpose of the present study was to measure the AT length under consideration of its curvature using a modified number of foil markers compared to our previously proposed method [10] during walking and running at different speeds. The error on the AT length and maximum strain introduced by a given marker number with respect to the length calculated with all markers was investigated by systematically eliminating markers from the curved length. We hypothesized significant inaccuracies in the AT tendon length measurement across gait phases, gait modes and gait speeds of the AT straight compared to AT curved approach. We further hypothesized that the AT length and strain could be determined with a tolerable error by a reduced marker-set, i.e., fewer markers than all available markers.

## 2. Materials and Methods

### 2.1. Experimental Design

In this study, 11 young, healthy individuals participated (age 28.0 ± 2.6 years, height 175.0 ± 7.1 cm, body mass 75.0 ± 12.0 kg). All participants gave written informed consent to the experimental procedure, which was approved by the ethics committee of the Humboldt-Universität zu Berlin (HU-KSBF-EK_2018_0005) and in accordance with the Declaration of Helsinki. After 10 min of familiarization, participants walked at 1.4 m/s and ran at 2.5 m/s (slow running) and 3.5 m/s (fast running) on a treadmill (Daum electronic, ergorun premium8, Fürth, Germany). During gaits, the length of the AT was defined as the distance between the origin on the GM MTJ and the insertion on the calcaneus bone, under consideration of the concave curvature of the AT in the longitudinal axis (Figure 1a). The insertion position was captured by a reflective foil marker placed on the notch of the calcaneus bone (Figure 1a). The curvature was assessed using several of the same reflective foil markers placed along the curved length of the AT on the skin (Figure 1a). The position of the GM MTJ was determined using ultrasound and a projection to the skin surface, which was then transformed to the global coordinate system to align insertion, foil, and origin markers (Figure 1a). For the simplification of the AT length determination with a reduced number of reflective foil markers, we then assessed the deviation of AT length that would be introduced when removing one or more reflective foil markers between origin and insertion by calculating the absolute length error compared to the reference criterion, i.e., the AT length determined by all reflective markers. Furthermore, we calculated the straight length of the AT as a Euclidean distance between AT origin and insertion.

### 2.2. Gait Event Detection

During walking, the touchdown of the right foot was determined as the minimal vertical position of the heel marker [26] and during running as the first maximum extension of the knee angle [27]. The foot-off was defined as the reversal of the anterior–posterior velocity of the toe marker during walking and as the second maximum extension of the knee angle during running [28]. For this purpose, the right leg ankle and knee joint kinematics were assessed by six reflective markers placed on the tip of the toe, medial and lateral malleolus, medial and lateral epicondyle of the femur, and the greater trochanter. The marker trajectories were filtered with a fourth-order, low pass, and zero-phase shift Butterworth filter with a cutoff frequency of 12 Hz.

### 2.3. Achilles Tendon Length and Strain Assessment

For the AT length determination during gaits, a reflective marker-based motion capture approach was implemented and combined with ultrasonography [10]. The position of the AT insertion on the notch of the calcaneus bone (identified by means of ultrasound) as well as the curvature of the AT between origin and insertion were captured using a line of reflective foil markers placed on the skin (distal marker on calcaneus notch, 20 mm intervals, 5 mm in diameter) using a motion capture system (Vicon Motion Systems, Oxford, UK) integrating 14 cameras at a sampling frequency of 250 Hz. The GM MTJ, as the AT origin, was defined as the most distal ending of the GM and tracked during gaits using a 6 cm linear ultrasound transducer operating at 146 Hz (Aloka UST-5713T, Hitachi Prosound, alpha 7, Japan), which was aligned with the reflective foil marker path (Figure 1a). The GM MTJ was tracked using a semi-automatic tracking algorithm from the stack of the US images [10]. The position of the GM MTJ was then transferred to the skin surface, which was detected by using the ‘Canny edge detection algorithm’ [29] in the same actual ultrasound image. The transformation of GM MTJ to the skin surface was done as the shortest distance of the GM MTJ to skin (Figure 1a). The position and the orientation of the ultrasound transducer in 3D space were captured employing a mounted custom marker tripod and the motion capture system (Figure 1a). For the transformation of the defined coordinate system in the ultrasound image to the global coordinate system, a custom-made calibration tool was used in order to digitize the four corners of the protective front layer of the ultrasound transducer and then a coordinate system was defined on the left-center of the front protective layer. During a separate static preparation trial, the position of the tripod on the probe was determined relative to the defined coordinate system on the front protective layer of the transducer. The AT length was then calculated as the sum of Euclidean distances between all foil markers from the insertion to the origin (GM MTJ). The averaged AT length from all participants, calculated by all possible reflective foil markers along the AT curve during walking, slow running and fast running over the gait cycle, is shown in Figure 1b. A manual trigger was used to synchronize the ultrasound device and the motion capture system. Ten complete gait cycles in each gait speed per individual were captured for the kinematic and AT length assessments and used for further analysis. The entire methodology, including the GM MTJ projection to motion capture coordination system and validation of the semi-automatic tracking algorithm, has been described in detail previously [10]. Furthermore, the AT strain during walking and running was calculated by dividing the instantaneous difference of AT length and AT resting length, which was measured at rest in 20${}^{{}^{\circ}}$ plantar flexion, where tendon slackness has been reported previously [20].

### 2.4. Reduction of Reflective Foil Marker Number for Achilles Tendon Curvature Determination

The number of reflective foil markers on the AT path (i.e., all reflective markers excluding the insertion and the origin of the AT) to assess its curvature varied because of the individuals’ GM MTJ position, which could be more or less distal on the shank (Figure 1a). The individuals with the same number of foil markers were clustered, i.e., P5: individuals with five foil markers (two participants), P6: six foil markers (two participants), P7: seven foil markers (four participants), P8: eight foil markers (two participants), and P9: nine foil makers (one participant).

The AT length was first calculated without any foil marker as a straight length, connecting the origin (GM MTJ) and insertion (calcaneus marker). The AT length was also calculated with all foil markers as the reference criterion (i.e., reference length). Then the number of foil markers was systematically reduced one by one, where MS8 (marker-set) to MS1 indicates the number of used foil markers for each participant. The AT length in each of the available marker-sets was calculated considering all possible combinations (i.e., the order does not matter) of the foil makers (i.e., different markers positioned on the AT curved path) using Equation (Equation 1):(1)$$nc_{ji}=\frac{n_{j}!}{\left(n_{j}-r_{i}\right)!\times r_{i}!}$$
where *j* is the index of the participant, *i* refers to the marker-set (i.e., MS1-8), nc is the total number of combinations of the different foil marker positions, *n* is the total number of foil markers, and *r* is the selected number of foil markers depending on the used marker-set (i.e., from 1 to 8). The calculated AT length in each individuals’ combination of each marker-set ($nc_{i}$) over the whole stride was compared to the reference length by means of the root mean square error (RMSE). The combination with the lowest RMSE was selected as the optimal combination for the selected marker-set for each participant and expressed for further analysis as absolute AT length error, i.e., the AT length difference to the reference curved length in mm.

Further, we analyzed the error of the foil marker reduction on the AT maximum strain with respect to the AT maximum strain calculated with all foil markers in order to analyze the error effect associated with length changes of the AT during gaits normalized to the individual AT resting length.

### 2.5. Statistics

A Wilcoxon signed-rank test was used to test for differences between AT length and maximum strain determined by either the curved length approach or straight length approach, including all gait speeds and phases. A linear mixed model was conducted to test for the main effects of marker-set (MS1-8), gait speed (walking, slow and fast running) and gait phase (swing vs. stance) on the phase-averaged AT length error and AT maximum strain error. In case of a significant interaction effect, a post hoc analysis was performed, and Benjamini-Hochberg corrected *p*-values will be reported. The normal distribution of the linear mixed model residuals of the AT length and maximum strain error was tested using the Shapiro-Wilk test and revealed non-normal distribution (p<0.001). However, since linear mixed models are robust against violations of this assumption, we adhered to this concept [30]. The significance level was set to α=0.05, and all values are reported as means and standard deviation. The statistical analyses were conducted using R v4.0.1 (R foundation for statistical computing, Vienna, Austria. Packages, ‘nlme package’ was used for the linear mixed model and ‘emmeanse’ was used for post hoc testing). To assess at which percentage of all used markers the measurement accuracy is not substantially decreased by removing further markers, the absolute length difference averaged over both phases between the AT curved length and AT length including a reduced relative foil marker number were calculated. The AT length measurement error of each individual was interpolated linearly to 100 points, and then a linear regression was fitted to all data points in 1% intervals starting from 0% (i.e., straight length) to 100% (i.e., reference length). The resultant $R^{2}$ values of each interval were then calculated in each step. The percentage of remaining markers where the average $R^{2}$ curve saturated was used as an indicator of a sufficient marker number, i.e., the addition of more markers would not significantly improve the accuracy. For this specific percentage of remaining markers, we calculated the respective locations of the foil markers as an average of the individual best marker combination (lowest RMSE) across phases and participants, to provide a recommendation for the application of a reduced marker-set. The positions were expressed as the percentage of the AT length during upright standing relative to the calcaneus marker (i.e., 0%). The length-error for each individual that is introduced when applying the average marker position and not the individual optimal position was also calculated as length-difference (RMSE over the whole stride).

## 3. Results

The reference AT length throughout the entire gait cycle ranged between 197 ± 25 mm and 210 ± 27 mm during walking; 199 ± 26 mm and 210 ± 27 mm during slow running; and 199 ± 25 mm and 211 ± 27 mm during fast running (Figure 1b). On average, the curved reference length of the AT was significantly longer than the straight length across all gait speeds and gait phases with individual maximum differences of up to 4.3 ± 1.0 mm during walking, 5.4 ± 1.1 mm during slow running, and 5.7 ± 1.2 mm during fast running (Figure 2).

The AT length measurement error for each marker-set with respect to the reference length during the entire gait cycle of walking, slow running and fast running is illustrated in Figure 2. In all three gait speeds, the AT length measurement error was larger for marker-sets with fewer foil markers compared to those with a greater number over the entire gait cycle. Increased AT length measurement errors were observed at the initial stance phase and during the first half of the swing phase in all three gait speeds, yet most prominent in MS1 and MS2 (Figure 2). For the phase-averaged AT length measurement error, the linear mixed model revealed a significant main effect of marker-set (p<0.001) and gait phase (p<0.001) but not of gait speed (p=0.752, Figure 3a). The swing phase-averaged AT length measurement error was larger than the stance phase-averaged error (Figure 3a,b).

The analysis also showed a significant interaction effect of marker-set and gait phase (p<0.0001). The post hoc analysis for the comparison of marker-set between swing and stance phases revealed significantly lower errors for MS1-3 (p<0.001,p<0.001, and p=0.012, respectively) during the stance phase compared to the swing phase and no significant differences between phases for MS4-8 (p=0.126,0.443,0.665,0.777; Figure 3a,b).

The AT length measurement error averaged over the entire gait cycle for the calculated percentage of remaining markers, and the corresponding $R^{2}$ is shown in Figure 4a,b. High inter-individual variability was obtained at values below 30% of the remaining markers (Figure 4a). Qualitatively, the $R^{2}$ reached and maintained a plateau at a value of 30% of markers for walking and 50% for running (Figure 4b). The average error of the AT length measurement was below 0.13 mm at 30% and 0.06 mm at 50% of the full marker-set in walking and running, respectively (Figure 4a).

The maximum AT strain occurred during the stance phase and was in the magnitude of 4.0 ± 1.2% during walking, 4.5 ± 1.4% during slow running and 4.9 ± 1.2% during fast running. AT maximum strain measured by the curved length method was significantly lower by 0.76 ± 0.17% than the straight AT length method across all gait speeds and phases (p<0.001). The AT maximum strain measurement error showed a significant main effect of marker-set (p<0.001), indicating that the AT maximum strain measurement error was higher with a reduced number of foil markers (from MS1 to MS8, Figure 5). No further significant main or interaction effects were found for the AT maximum strain measurement error (gait speed p=0.443, gait speed by marker-set interaction p=1.000). The maximum strain measurement error when using 30% of the markers was below 0.15% for walking and below 0.08% with 50% of the markers during both slow and fast running.

The average location of the foil markers for marker-set 2 to 4 is presented in Table 1. The averaged overall error associated when using this reduced average marker positions across participants and gait phases was for marker-set 2: walking 0.26 mm, slow running 0.28 mm, and fast running 0.29 mm; marker-set 3: walking 0.18 mm, slow running 0.17 mm, and fast running 0.18 mm; and marker-set 4: walking 0.15 mm, slow running 0.14 mm, and fast running 0.14 mm.

## 4. Discussion

Here we investigated the reconstruction of the in vivo AT curvature during locomotion using a various number of reflective foil markers on the line connecting the AT insertion and origin. We found a significantly longer length of the AT measured with the reference curved length approach compared to the straight length approach. The AT length and maximum strain measurement error decreased in marker-sets with more reflective markers. When using a minimalistic marker-set with only 30% of the markers of the reference marker-set during walking and 50% during running, the involved measurement error of AT maximum length and maximum strain were lower than 0.35 mm and 0.15% strain, respectively. The results suggest the feasibility of the reduced marker-set that involved only a marginal error for the in vivo AT length and strain determination during locomotion.

It has previously been recognized [20] that the AT curvature changes during contractions of the triceps surae muscles, ankle joint rotations, and different functional movement tasks [18,19] and, thus, need to be considered for an accurate AT length determination. For instance, Fukutani et al. [22] reported a maximum difference of 9.0 ± 5.4 mm between curved and straight AT length during passive ankle joint rotations. Furthermore, it has been shown that the AT curved length and strain during hopping differed by 5.0 ± 1.3 mm and 1.3 ± 0.7%, respectively, compared to the straight length and strain [23]. In agreement with the earlier studies, we found a significant difference of the straight length and maximum strain to the curved length and maximum strain of the AT over the three gait speeds and both gait phases. The straight length approach underestimated the AT length and showed a high individual variability in the resultant error. In our investigated participants, the observed maximum differences between the curved and straight length assessment ranged from 0.1 to 5.7 mm, i.e., 0.2% to 1.6% strain. Considering that the AT maximum strain during the stance phase was 4.0–4.9% across gait modes (walking and running), our results evidence that the straight length method may introduce important artifacts (5% to 40%) in maximum tendon strain measurements during walking and running.

The reconstruction of the AT curvature by an increasing number of foil markers continuously reduced the average and maximum AT length and maximum strain error. The findings showed that when including already 30% of the maximum available markers for walking and 50% for running, the $R^{2}$ of the AT length measurement error reached and maintained a very high value ($R^{2}$ > 0.99). Furthermore, the achieved steady-state in $R^{2}$ shows that a higher percentage of marker number (above the mentioned thresholds) would not significantly increase the measurement accuracy. The remaining errors in the maximum AT length and strain measurement were for walking 0.35 ± 0.11 mm and 0.15 ± 0.06%, for slow running 0.15 ± 0.08 mm and 0.06 ± 0.04%, and for fast running 0.17 ± 0.04 mm and 0.08 ± 0.02%.

The magnitude of these errors is arguably negligible when considering a given average AT length change of 7.8 ± 2.6 mm and average maximum strains between 4.0% and 4.9% across speeds and phases. Therefore, for the investigated participants and depending on the individual AT length, 2 to 4 markers between the origin and insertion of the AT are sufficient to determine the AT curvature across speeds and gait phases. Using the reduced marker-set instead of all markers would significantly simplify the assessment protocol in terms of cameras needed for marker capturing and post-processing and thus improve the feasibility of the method. For practical application purposes and based on the present results, Table 1 provides the position of markers in MS2 to MS4 (averaged among three gait speeds and two gait phases), expressed as relative positions with respect to the individual AT length during upright standing. According to a given individual AT length, an appropriate marker-set for a simplified assessment protocol can be chosen from Table 1. The presented average positions for markers in Table 1 do not affect the individual variability of the participants because there was no significant difference between marker positions in the different gait speeds (p=0.674) and gait phases (p=0.723, tested using a repeated-measures ANOVA). In addition, the final associated overall error when applying the average markers positions of the reduced marker-sets 2 to 4 was 0.20 ± 0.05 mm on average similarly during walking, slow running and fast running.

The AT length measurement error was significantly greater during the swing phase than the stance phase, indicated by the significant main effect, but the magnitude was rather small. Furthermore, this difference diminished at marker-sets with a greater foil marker number (<MS4). It is reasonable to assume that the greater error with fewer markers during swing arises from the high plantar flexion angles after toe-off, which affects the AT curvature. The gait speed showed no significant effect on the AT length measurement error, indicating a comparable curvature behavior between speeds and gait modes, suggesting that the same marker-set can be used for different gait speeds and modes.

## 5. Conclusions

In conclusion, we investigated the effect of foil marker reduction on the reconstructed AT curvature. Our results indicate that reducing the number of foil markers by 70% during walking and 50% during running would result in a marginal error and thus a negligible effect on the AT length and maximum strain measurement across gait phases and gait speeds. Therefore, a reduced marker-set can be recommended for future applications. The marker positions for each marker-set suggested here can be used to choose an appropriate marker-set for the individual AT length in a simplified assessment protocol.

## Figures and Tables

**Figure 1 sensors-21-07387-f001:**
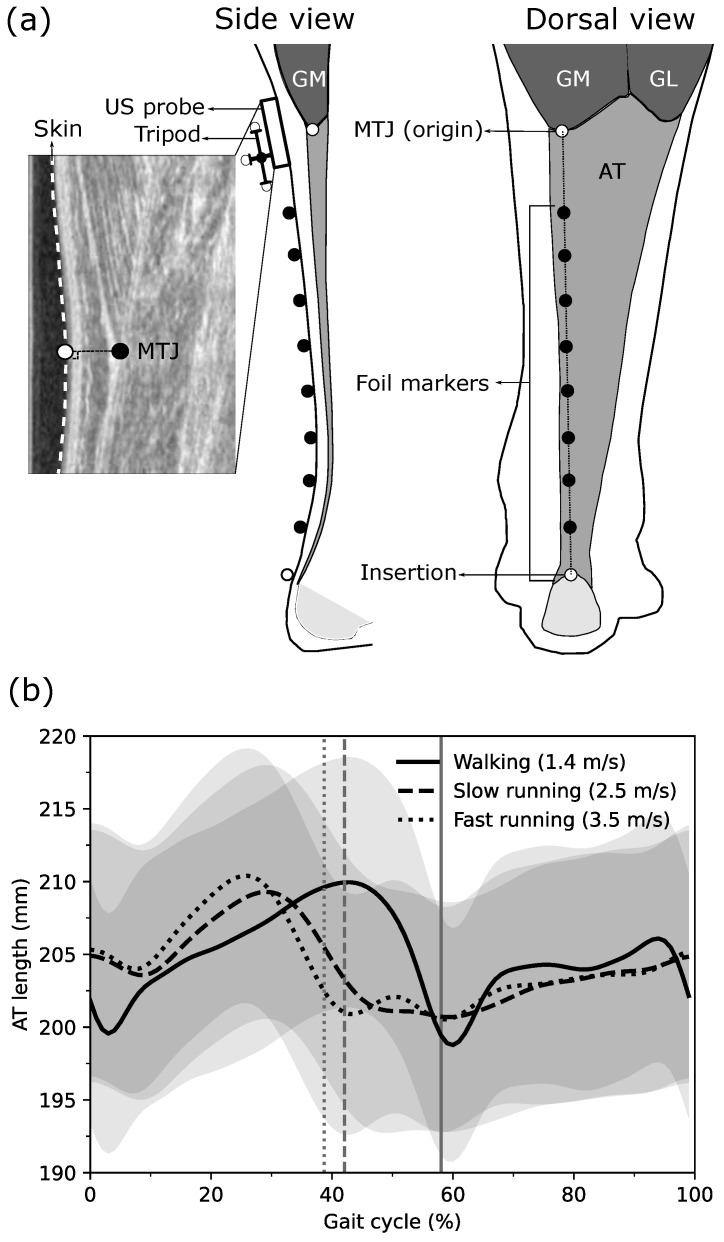
(**a**) Experimental setup for determining the Achilles tendon (AT) length during walking and running. Reflective foil markers on the skin were used to reconstruct the curvature of the AT (black markers) from origin to insertion (white markers). The gastrocnemius medialis (GM) myotendinous junction (MTJ), as the AT origin, was projected to the skin surface, and the coordinates of the ultrasound images were transferred to the global coordinate system using a tripod that was mounted on the ultrasound probe. (**b**) The length of the AT with all foil makers during walking, slow running and fast running throughout the gait cycle. Vertical lines separate stance and swing phases. GL: gastrocnemius lateralis, US: Ultrasound.

**Figure 2 sensors-21-07387-f002:**
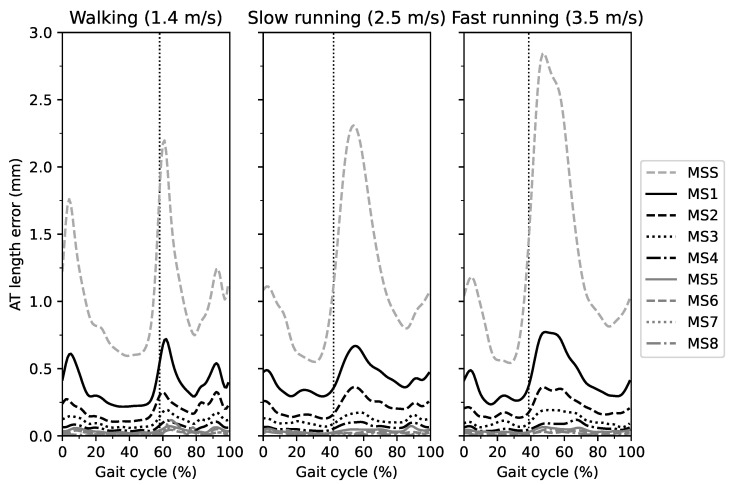
The absolute error of the Achilles tendon (AT) length measurement as the difference between the different marker-sets (MS) in comparison to the curved AT length (reference value) throughout the gait cycle. MSS is the straight AT length, and MS1-8 indicates the marker-set with one to eight foil markers. The vertical dashed line separates the stance and swing phase.

**Figure 3 sensors-21-07387-f003:**
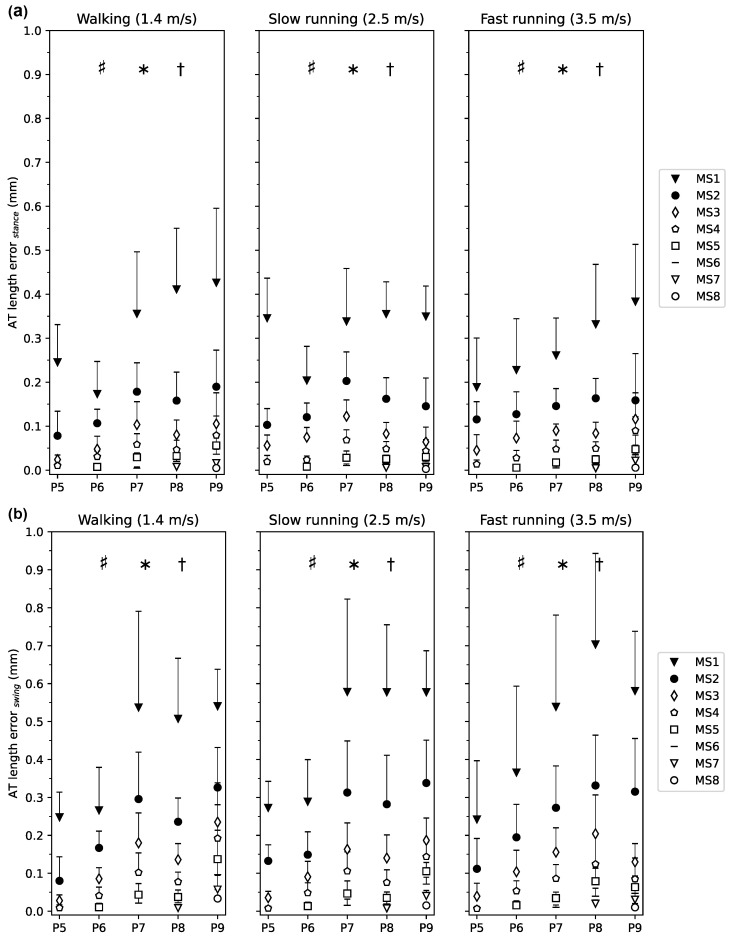
Averaged Achilles tendon (AT) length measurement error for each marker-set with respect to the curved reference length of the AT during the (**a**) stance phase and (**b**) swing phase of walking, slow and fast running. The marker-set with i number of foil markers (MSi) is the AT length with i = 1 to 8 foil markers. P5 to P9 are the clustered participants with the same number of foil markers. The error bars express the standard deviation of the AT length error throughout the phase (stance, swing), showing the variance over each phase and then averaged among individuals within each cluster. * main effect of phase (p<0.001), † main effect of marker-set (p<0.001), # interaction effect of phase and marker-set (p<0.001). The post hoc analysis for the comparison of marker-set between swing and stance phases revealed significant lower errors for MS1-3 (p<0.001,p<0.0001 and p=0.012, respectively) during the stance phase and no significant phase differences for MS4-8 (p=0.126,0.443,0.665,0.777).

**Figure 4 sensors-21-07387-f004:**
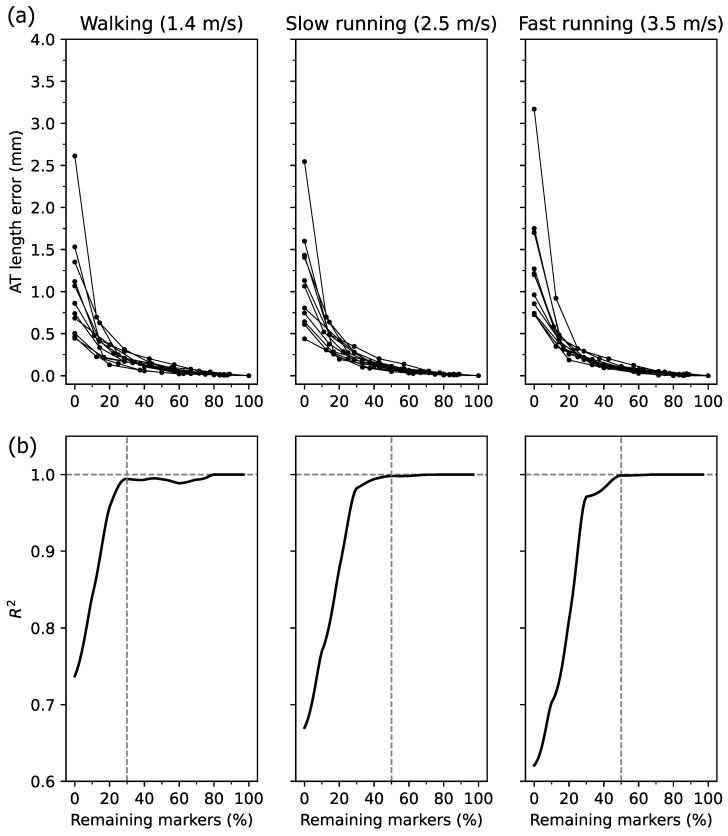
(**a**) Individual Achilles tendon (AT) length measurement error with respect to the reference length (i.e., the AT length with all foil makers) as a function of relative marker number. The 0% indicates the straight AT length, and 100% the curved reference AT length. Note the different number of markers between individuals. (**b**) The resultant $R^{2}$ of the linear regression on the data presented in panel (**a**) in % intervals from 0% to 100% of the remaining markers.

**Figure 5 sensors-21-07387-f005:**
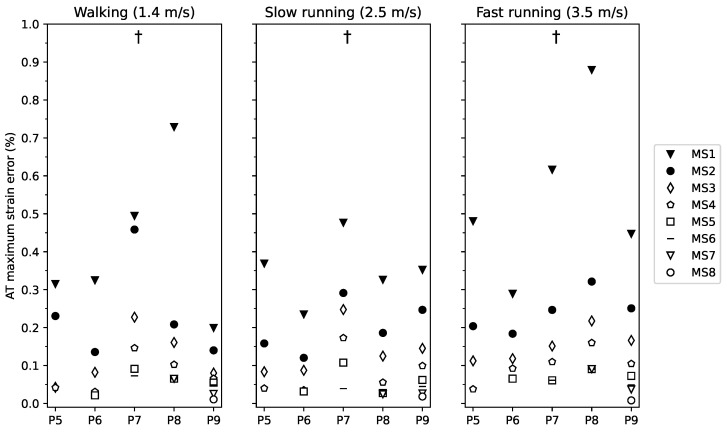
Maximum strain error of the Achilles tendon (AT) during walking, slow and fast running for marker-set 1-8 (markers-set (MS) with 1 to 8 foil markers) with respect to the marker-set with all possible foil reflective markers. Participants were clustered into five groups of P5-9 (participants with 5 foil makers to participants with 9 foil markers). † main effect of marker-set (p<0.001).

**Table 1 sensors-21-07387-t001:** Location of foil markers in the percentage of Achilles tendon (AT) length during quite upright standing relative to the AT insertion marker on the calcaneus bone for each marker-set. The locations refer to the reduced marker-set (i.e., 50% of all markers according to the individual AT length, thus, 2 to 4 markers) across the three gait speeds (1.4, 2.5, and 3.5 m/s) and two phases (stance and swing).

	Foil Marker 1 (%)	Foil Marker 2 (%)	Foil Marker 3 (%)	Foil Marker 4 (%)
Marker-set 2	22	56		
Marker-set 3	15	35	65	
Marker-set 4	10	31	47	66

## Data Availability

All data are available upon request.

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
