# Peer review of "A Simplified Method for Considering Achilles Tendon Curvature in the Assessment of Tendon Elongation"

_sensors, 2021, doi:10.3390/s21217387_

Round 1

Reviewer 1 Report

General comments

I have one general comment about the results.  Given the aim of the work the results presented directly support the conclusions drawn apart from the final table presented in the discussion.  It may be that I have missed it in the methods and results and although I agree with the results presented, I think something extra is needed to help support the final table values.  This would also move this into the results section as I was pondering by the end of the results exactly how use of the many combinations was used.

It seems to be that all the combinations of markers for a given number of markers were calculated and I at first thought this would be what was presented with the means and SD to show a spread of errors per MSi but this does not seem to be the case.  The combination for a subject with a given set of markers with the smallest error was chosen as the value that was then used to represent that performance and the following results based on this are all fine and well done.  However, given that the final outcome, and suggestion, is to pick markers at percentages of distances given the marker set number wouldn’t it be better to see the spread of error per marker number for all the combinations.  That is for say 6 markers from 9 then it may be that the best is much better any 5 from 9 but the worst 6 markers from 9, say all at one end, would be worse than the best 5 from 9.  This would also give a much better set of information as to what spread of markers gives the smaller errors as well as the number.  It would also make use of the full combinations data you have gone through the trouble of calculating.

Specific comments

You say foil makers and this seems a non-standard way to refer to what I eventually decided were just standard 5 mm diameter near spherical retroreflective markers.  Foil when referring to a material is a thin malleable metal.  I had though a true flat metal foil was being used and would be utilised with the ultrasound as well as Vicon. Could you clarify please what the markers were.

Line 167

Within the sentence where RMSE defined explicitly state it is over the stride, as you have figures where it is split between swing and ground contact.

Table 1 is a results table and should be in the results, along with how it was calculated in the methods.

Author Response

Reviewer 1

General comments

I have one general comment about the results. Given the aim of the work the results presented directly support the conclusions drawn apart from the final table presented in the discussion. It may be that I have missed it in the methods and results and although I agree with the results presented, I think something extra is needed to help support the final table values. This would also move this into the results section as I was pondering by the end of the results exactly how use of the many combinations was used. It seems to be that all the combinations of markers for a given number of markers were calculated and I at first thought this would be what was presented with the means and SD to show a spread of errors per MSi but this does not seem to be the case. The combination for a subject with a given set of markers with the smallest error was chosen as the value that was then used to represent that performance and the following results based on this are all fine and well done. However, given that the final outcome, and suggestion, is to pick markers at percentages of distances given the marker set number wouldn't it be better to see the spread of error per marker number for all the combinations. That is for say 6 markers from 9 then it may be that the best is much better any 5 from 9 but the worst 6 markers from 9, say all at one end, would be worse than the best 5 from 9. This would also give a much better set of information as to what spread of markers gives the smaller errors as well as the number.  It would also make use of the full combinations data you have gone through the trouble of calculating.

Response:

Thank you for your valuable review and this general comment. It is correct that the combination with the smallest RMSE was chosen for the analysis to account for the inter-individual variability. In the appendix to this response letter, we provide a table of all the AT length error data for each participant and every combination in each marker-set separately for all speeds (stride average) as requested by the reviewer (file name:All combinations_speed.xlsx’, ‘excel file description.txt’). Note that the values are presented on an individual basis because the best combination for each marker-set could have been different between participants in each cluster (i.e., participants with the same number of markers). Further and as suggested by the reviewer, a second table is provided (file name: Averaged errror.xlsx), showing the mean and standard deviations of the AT length error between marker-sets, participants and gait speeds based on the data presented in the first table, to indicate the spread of AT length error.

However, because of the inter-individual variability (the best combination is different between participants in each marker-set), we think it is not appropriate and helpful to provide all those data for the reader and would rather stay with our analysis approach. To address the reviewers’ comment, we calculated the standard deviation of the individual marker positions along with the already presented average position values shown in the final table for marker-set 2 to 4 (see table below). We then re-calculated the AT length error for each individual over a stride that is introduced when applying the average marker position. The results show that although there is some deviation of the marker position (SD in the table), the associated average error was rather small, i.e., marker-set 2: walking 0.26 mm, slow running 0.28mm and fast running 0.29 mm; marker-set 3: walking .018 mm, slow running 0.17 mm and fast running 0.18 mm; marker-set 4: walking 0.15 mm, slow running 0.14 mm and fast running 0.14mm.

Table 1. Location of foil markers in the percentage of Achilles tendon (AT) length during quite upright standing relative to the AT insertion marker on the calcaneus bone for each marker-set. The locations refer to the reduced marker-set (i.e., 50% of all markers according to the individual AT length, thus, 2 to 4 markers) across the three gait speeds (1.4, 2.5 and 3.5 m/s) and two phases (stance and swing).

Foil marker 1 (%)

Foil marker 2 (%)

Foil marker 3 (%)

               Foil marker 4 (%)

Marker-set 2

22±13

56±13

Marker-set 3

15±09

35±14

65±06

Marker-set 4

10±05

31±10

47±08

66±06

According to the reviewers' comment, we moved the table to the results section and added further information on the calculation to the methods and results sections, as well as discussed this aspect in the discussion of the revised version as follows.

In the methods section:

Line 199-209: " The percentage of remaining markers where the average R2 curve saturated was used as an indicator of a sufficient marker number, i.e., the addition of more markers would not significantly improve the accuracy. For this specific percentage of remaining markers, we calculated the respective locations of the foil markers as an average of the individual best marker combination (lowest RMSE) across phases, participants but not in gait speeds to provide a recommendation for the application of a reduced marker-set. The positions were expressed as the percentage of the AT length during upright standing relative to the calcaneus marker (i.e., 0%). In addition, the length-error for each individual that is introduced when applying the average marker position and not the individual optimal position was also calculated as length-difference (RMSE over whole stride).”

In the results section:

Line 256-261: " The average location of the foil markers for marker-set 2 to 4 is presented in Table 1. The averaged overall error associated when using this reduced average marker positions across participants was for marker-set 2: walking 0.26 mm, slow running 0.28 mm and fast running 0.29 mm; marker-set 3: walking 0.18 mm, slow running 0.17 mm and fast running 0.18 mm and marker-set 4: walking 0.15 mm, slow running 0.14 mm and fast running 0.14 mm.

In the discussion section:

Line 315-318:” In addition, the final associated overall error when applying the average markers positions of the reduced marker-sets 2 to 4 was 0.20±0.05 mm in average during walking, slow running and fast running.

Specific comments

You say foil makers and this seems a non-standard way to refer to what I eventually decided were just standard 5 mm diameter near spherical retroreflective markers.  Foil when referring to a material is a thin malleable metal.  I had though a true flat metal foil was being used and would be utilized with the ultrasound as well as Vicon. Could you clarify please what the markers were.

Response:

Thanks for this comment. The foil markers were self-adhesive retro-reflective tape circles 5 mm in diameter (see photo below).

The following definition has been added to the manuscript (line 41):

"Foil markers (i.e., self-adhesive retro-reflective tape circles with 5 mm in diameter) …"

Comment on L167:

Within the sentence where RMSE defined explicitly state it is over the stride, as you have figures where it is split between swing and ground contact.

Response:

Thank you for this comment. The following sentence has been changed in the revised manuscript (line 167):

"The calculated AT length in each individuals' combination of each marker-set (nci) over the whole stride was compared to the reference length by means of the root mean square error (RMSE)"

Comment:

Table 1 is a results table and should be in the results, along with how it was calculated in the methods.

Response:

Thank you for this comment. We have relocated the table to the results section and added further information on the calculation to the method and results sections (see also the response to the general comment).

Reviewer 2

The study is a very careful, and warranted, examination of the influence of marker set number on estimations of Achilles tendon length during gait. Achilles tendon curvature is not typically accounted for during gait analysis. The authors are congratulated in pushing the field forwards through their experiments in this area.

Response: Thank you for your review.

Reviewer 2 Report

The study is a very careful, and warranted, examination of the influence of marker set number on estimations of Achilles tendon length during gait. Achilles tendon curvature is not typically accounted for during gait analysis. The authors are congratulated in pushing the field forwards through their experiments in this area.

Author Response

Reviewer 2

The study is a very careful, and warranted, examination of the influence of marker set number on estimations of Achilles tendon length during gait. Achilles tendon curvature is not typically accounted for during gait analysis. The authors are congratulated in pushing the field forwards through their experiments in this area.

Response: Thank you for your review.
